# The Effects of TGF-β Signaling on Cancer Cells and Cancer Stem Cells in the Bone Microenvironment

**DOI:** 10.3390/ijms20205117

**Published:** 2019-10-15

**Authors:** Mitsuru Futakuchi, Kris Lami, Yuri Tachibana, Yukari Yamamoto, Masahiro Furukawa, Junya Fukuoka

**Affiliations:** 1Department of Pathology, Nagasaki University Hospital, Nagasaki 852-8501, Japan; mfuy.lopef@yahoo.ne.jp (Y.Y.); fukuokaj@nagasaki-u.ac.jp (J.F.); 2Department of Pathology, Nagasaki University Graduate School of Biomedical Sciences, Nagasaki 852-8523, Japan; krislami839@gmail.com (K.L.); y.tachibana0221@gmail.com (Y.T.);; 3Department of Molecular Toxicology, Graduate School of Medical Sciences, Nagoya City University, Nagoya 467-8601, Japan

**Keywords:** bone metastasis, mammary tumor, cancer stem cell, bone microenvironment, TGF-β, non CSC

## Abstract

Background: Transforming growth factor-β (TGF-β) plays a key role in bone metastasis formation; we hypothesized the possible involvement of TGF-β in the induction of cancer stem cells (CSCs) in the bone microenvironment (micro-E), which may be responsible for chemo-resistance. Methods: Mouse mammary tumor cells were implanted under the dorsal skin flap over the calvaria and into a subcutaneous (subQ) lesions in female mice, generating tumors in the bone and subQ micro-Es. After implantation of the tumor cells, mice were treated with a TGF-β R1 kinase inhibitor (R1-Ki). Results: Treatment with R1-Ki decreased tumor volume and cell proliferation in the bone micro-E, but not in the subQ micro-E. R1-Ki treatment did not affect the induction of necrosis or apoptosis in either bone or subQ micro-E. The number of cells positive for the CSC markers, SOX2, and CD166 in the bone micro-E, were significantly higher than those in the subQ micro-E. R1-Ki treatment significantly decreased the number of CSC marker positive cells in the bone micro-E but not in the subQ micro-E. TGF-β activation of the MAPK/ERK and AKT pathways was the underlying mechanism of cell proliferation in the bone micro-E. BMP signaling did not play a role in cell proliferation in either micro-E. Conclusion: Our results indicated that the bone micro-E is a key niche for CSC generation, and TGF-β signaling has important roles in generating CSCs and tumor cell proliferation in the bone micro-E. Therefore, it is critically important to evaluate responses to chemotherapeutic agents on both cancer stem cells and proliferating tumor cells in different tumor microenvironments in vivo.

## 1. Introduction

Breast cancer has the highest incidence among all malignancies and is the leading cause of death among women worldwide [1]. Bone is the preferential metastatic site for breast cancer regardless of the cancer subtype [2,3]. Bone metastasis leads to a number of complications grouped under the term: skeletal-related events [4]. Despite progress in the medical care and the novel therapeutics being developed, the fatality rate due to metastases remains high [5]. The 10-year survival rate of the patients with local recurrence of breast cancer is 56%, while those with metastatic diseases falls to 9% [6]. Once bone metastatic lesions become manifest, most of the lesions begin to exert resistance to the conventional chemotherapy [7].

Therapy resistance represents a significant hurdle in the treatment of breast cancer, forcing the development of alternative strategies. The bone microenvironment (micro-E) has been described as a fertile soil for metastasis, where conditions are favorable for the survival and proliferation of the tumor cells [8,9,10]. It is generally acknowledged that the interaction between the metastatic tumor cells and bone stromal cells in the bone micro-E provides a particularly fertile soil to promote aggressive behavior of the malignant tumor cells that arrived at bone micro-E [11]. Abundant cytokines and growth factors stored in the bone matrix would be released in the bone micro-E upon its resorption by osteoclasts, which promote the malignant behavior of tumor cells in an autocrine and/or paracrine fashion [12,13].

Because not all cancer cells in the in vivo tumor tissue have similar proliferating potentials, chemotherapeutic agents can substantially reduce the tumor volume by removing cancer cells with high proliferating potential, while having little effect on cancer cells with lower proliferation potential, such as cancer stem cells (CSCs), that form only a small fraction of the whole tumor. CSCs are a subpopulation of cancer cells that are able to self-renew and differentiate, and consequently, can be considered the seeds of cancers. Previously, we found that the number of CSCs which are positive for SOX2, CD44, and CD166, in the bone micro-E, was significantly higher than those in the subcutaneous (subQ) micro-E, indicating that the bone micro-E is a key niche for CSCs [14]. It is possible that CSCs in the bone micro-E may be responsible for chemo-resistance because of their slow growth rates [15]. Therefore, careful monitoring of both cancer cells and CSCs is required to evaluate responses to therapeutic agents.

Transforming growth factor-β (TGF-β) plays a key role in osteoclast induction as well as in the proliferation of metastatic tumor cells in the bone micro-E [11]. TGF-β released from the bone matrix upon its resorption by osteoclasts has also been demonstrated to increase the expression of the parathyroid thyroid hormone-related peptide (PTHrP). PTHrP stimulates osteoclast activation, which causes bone destruction. This bone destruction in turn releases factors that stimulate further bone destruction and promote tumor growth [8]. Previously, we demonstrated that TGF-β promoted the malignant potential of prostate cancer in rats [16] and mammary tumors in mice [17] in the bone micro-E.

In the present study, we hypothesized the possible involvement of TGF-β for the induction of CSCs in the bone micro-E. We used our animal model [14,16,17,18,19] to test this hypothesis and to also confirm the effect of TGF-β in the bone micro-E: mouse mammary tumor cell line, CL66M2 was implanted into two different sites in female mice, under the dorsal skin flap over the calvaria and into the subcutis of the back of each female mouse, generating tumor growth in the bone and subQ micro-Es. Because tumor tissue in vivo is heterogeneous [20], containing proliferating cancer cells and CSCs in each micro-E, the effect of TGF-β on proliferating cancer cells was evaluated by tumor volume, cell proliferation, and the induction of apoptosis and necrosis, and we also evaluated the effect of TGF-β on cancer stem cells. We also examined expression of ERK1/2, AKT, and BMP signaling in both micro-E.

## 2. Results

### 2.1. Suppression of TGF-β Signaling by a TGF-β Receptor 1 Kinase Inhibitor (R1-Ki) in the Bone Micro-E

To examine the effects of TGF-β on our tumor-bone invasion model, we implanted the mouse mammary tumor cell line CL66M2 into two different sites in female mice, under the dorsal skin flap over the calvaria and into a subQ lesion, and then injected the mice six times with TGF-β receptor 1 kinase inhibitor (R1-Ki) over the course of the experimental period (Figure 1A). In those mice, the bone micro-E (Figure 1B) and subQ micro-E (Figure 1C) can be easily delineated. Strong osteolysis associated with induction of numerous osteoclasts was observed in the bone micro-Es (Figure 1B). In the subQ micro-Es, the tumor cells grew with micro vessel invasion (Figure 1C).

To demonstrate the effects of TGF-β signal transduction, we examined TGF-β levels and the expression of phosphorylated SMAD2, which is a downstream molecule of TGFβ signaling. The level of TGF-β was significantly higher in the bone micro-E compared to the subQ micro-E. Treatment with R1-Ki did not significantly change TGF-β levels in either micro-E. Western blot analysis revealed that the expression of p-SMAD2 was high in the bone micro-E and low in the subQ micro-E (Figure 1D), and expression of p-SMAD2 in the bone micro-E was decreased by treatment with R1-Ki (Figure 1D). p-SMAD2 staining revealed a high number of positive cells in the bone micro-E in the control mice and a lower number of positive cells in the bone micro-E in the R1-Ki treated mice (Figure 1E,F). Quantitative analysis of p-SMAD2 positive cells showed that a significantly higher number of positive cells were present in the bone micro-E compared to the subQ micro-E, and that R1-Ki decreased the number of p-SMAD2 positive cells in the bone micro-E (Figure 1G). These results indicate that R1-Ki treatment significantly reduced TGF-β signaling in the tumor cells in the bone micro-E, but not in the subQ micro-E.

To confirm that the reduction of TGF-β signaling affects osteolysis and osteoclast in the tumor tissue in vivo, we evaluated the effect of R1-Ki on osteolysis and on osteoclast induction in the bone micro-E. Bone destruction was determined by the ratio of the area of bone destruction to the total area of the cranial bone (bone destruction index, Appendix A). Osteolysis was significantly decreased by R1-Ki treatment (Appendix A). Tartrate-Resistant Acid Phosphatase (TRAP) staining revealed a significantly higher number of osteoclasts in the bone micro-E in the control mice compared to the R1-Ki treated mice (Appendix A). These results confirmed the reduction of TGF-β signaling by R1-Ki treatment, and this reduction significantly decreased osteoclast induction and bone destruction in vivo.

### 2.2. The Effects of TGF-β on Tumor Growth and Cell Proliferation

In the bone micro-E, we observed an increased tumor growth in the control mice compared to the R1-Ki treated mice, resulting in a significant difference in tumor size on Day 24 (Figure 2A). The tumor grew more slowly in the subQ lesion compared to the growth in the bone lesion, and R1-Ki treatment did not suppress the tumor growth in the subQ micro-E (Figure 2B). In the bone micro-E, a significantly higher number of Ki-67 positive cells were observed in the control mice (Figure 2C). Treatment of R1-Ki significantly reduced the index of Ki-67 positive cells in the bone micro-E (Figure 2D,E), but not in the subQ micro-E (Figure 2E). These results indicate that TGF-β signaling is involved in tumor growth and the tumor cell’s proliferation in the bone micro-E, but not in the subQ micro-E.

### 2.3. The Effect of TGF-β Signaling on the Induction of the Necrotic Area and Apoptosis

Generally, the effectiveness of chemotherapeutic agents on a tumor is evaluated by the increase in the necrotic area in the tumor tissues. If the tumor is sensitive to chemotherapeutic agents, the necrotic area will increase, and if it is resistant, the necrotic area will not increase. The necrotic areas in the tumors were analyzed by microscopic analysis and image analyzer (Figure 3A,B). Quantitative analysis of the necrotic area in the tumor revealed that R1-Ki treatment did not change the proportion of the necrotic area in the bone micro-E (Figure 3C).

To investigate the effects of R1-Ki on the induction of apoptosis, we examined the expression of cleaved caspase 3 in the bone and subQ micro-Es in the control and treated mice. Immunohistochemical (IHC) staining of cleaved caspase 3 in the bone micro-E of control mice (Figure 4A) and R1-Ki treated mice (Figure 4B) and in the subQ micro-E of the control and treated mice revealed that R1-Ki treatment did not affect the number of positive cells in either the bone (Figure 4C) or subQ micro-E (Figure 4C). These results indicate that R1-Ki did not affect the induction of apoptosis in the tumor regardless of micro-E.

### 2.4. The Effect of TGF-β Signaling on the Induction of CSC in the Bone Micro-E

In this study we explored the possible involvement of TGF-β on the induction of CSC in the bone and subQ micro-Es, because we have previously demonstrated that bone micro-E is the key niche of CSC [14]. IHC study of SOX2 in the bone micro-E of control mice (Figure 5A) and R1-Ki treated mice (Figure 5B) revealed that R1-Ki treatment significantly reduced the number of SOX2 positive cells in the bone micro-E (Figure 5C). The number of SOX2 positive cells was significantly higher in the bone micro-E compared to the subQ micro-E (Figure 5C), and R1-Ki treatment significantly reduced the number of SOX2 positive cells in the bone micro-E but not in the subQ micro-E (Figure 5C). To confirm the suppressive effect of R1-Ki on the induction of CSC in the bone micro-E, we examined the number of CD166 positive cells in the bone microE in the control mice (Figure 5D), and similar results were obtained for CD166 positive cells (Figure 5D–F). These results indicate that the bone micro-E is a key niche for cancer stem cells, and that TGF-β is involved in the induction of CSC in the bone micro-E.

### 2.5. The Involvement of ERK1/2, AKT, and BMP Signaling in Tumor Growth within the Bone and subQ Micro-Es

Western blot analysis revealed that the expression of ERK 1/2 was similar in the bone and subQ micro-Es (Figure 6A). In contrast, the expression of phosphorylated-ERK1/2 (p-ERK 1/2) was high in the bone micro-E and low in the subQ micro-E, and the expression of p-ERK 1/2 in the bone micro-E was decreased by treatment with R1-Ki but did not affect in the subQ micro-E (Figure 6A). IHC staining of p-ERK1/2 in the bone micro-E in the control mice (Figure 6B) and R1-Ki treated mice (Figure 6C) revealed that R1-Ki treatment significantly reduced the number of p-ERK1/2 positive cells in the bone micro-E (Figure 6D). The number of p-ERK 1/2 positive cells was significantly higher in the bone micro-E compared to the subQ micro-E, and R1-Ki treatment significantly reduced the number of positive cells in the bone micro-E, but not in the subQ micro-E (Figure 6E). To examine the involvement of BMP signaling, we examined the expression of SMAD1/5, which is a downstream molecule of BMP signaling. There was only a very low level of SMAD1/5 detected, and p-SMAD1/5 was not detected in either the bone or subQ micro-Es, regardless of treatment (Figure 6E). Expression of p-SMAD1/5 and SMAD1/5 was confirmed by western blot analysis of extracts from BMP-4 treated HeLa cells and untreated HeLa cells, respectively (Appendix A).

These results suggest the involvement of ERK 1/2, and AKT signaling, but not BMP signaling in the tumor growth in the bone micro-E, and that ERK1/2, AKT, and BMP are not active in the subQ micro-E.

## 3. Discussion

Bone metastasis consists of multiple processes, that involves invasion, survival in the circulation, arrest in bone, and re-growth in the bone micro-E [21]. Ideally, an in vivo model of bone metastases should reflect the natural course of the disease and accurately simulate the disease progression commonly observed in advanced cancer patients. However, such an animal model that fully reflects the biology of bone metastasis is a major obstacle to developing therapies [22,23,24], because each process of bone metastasis involves an independent mechanism. We have developed a rat bone invasion model to observe the growth of prostate or breast cancer cells with osteoblastic/osteolytic lesions in the bone micro-E [14,16,17,18,19]. Although our models do not recapitulate the entire metastatic process that primary tumor cells in prostate or breast move to the typical location in the bone, our model does represent how metastatic tumor cells interact with the bone stromal cells in the bone microenvironment. Using our model, we identified a unique pattern of gene expression that was up-regulated at the tumor bone interface, including genes such as *RANKL*, which are known to be involved in bone metastasis [17,18]. Therefore, our model provides an exciting opportunity to elucidate the molecular mechanisms and to develop therapeutic targets underlying tumor stromal interaction in the bone micro-E.

In the previous study, we examined the effects of TGF-βRI kinase inhibitor on the expression of p-SMAD2, tumor size, and cell proliferation [17], and we compared the tumor cells at the bone-interface with those in the area far away from the bone but within the same tumor. Because those two areas are located in the same tumor, we could not completely deny the possibility that tumor cells in the area far away from the bone are affected by TGF-β at the bone-interface. In addition, the definition of tumor-bone interface and tumor-bone micro-E remained obscure. Therefore, in this study, we defined the bone microE as the area where TGF-β has influence on because the purpose of this study was the effects of TGF-β signaling in the bone micro-E.

TGF-β was shown to stimulate osteoclast induction and differentiation [25,26]. Using an antisense nucleotide oligo (ASO) of TGF-β and a neutralizing antibody, we previously demonstrated that the involvement of TGF-β in osteoclast induction was associated with the growth of prostate cancer and breast cancer in the bone micro-E [16,17], and proposed TGF-β signaling as a therapeutic target. In the present study, we demonstrated that treatment with a TGF-β R1-Ki (CAS 396129-53-6) significantly reduced the expression of phospho-SMAD2 (an indicator of the TGF-β signal transduction [27]), which reduced osteoclast induction and osteolysis, and suppressed tumor cell proliferation in the bone micro-E. Importantly, inhibition of TGF-β signaling leads to the suppression of the induction of CSCs in the bone micro-E. An additional finding of the present study was that R1-Ki was not effective in the subQ micro-E, indicating that preclinical testing of potential therapies for metastatic disease in animal models should include effects on both CSC induction and tumor cell proliferation in different metastatic niches.

In in vivo tumor tissue, cancer cells are heterogeneous in terms of nutrient availability [28], their potential for proliferation, and their potential for differentiation. The tumor micro-E has been demonstrated to be involved in the spatiotemporal dynamics of the cancer cells, including abundant phenotypic and functional heterogeneity [29,30]. We have demonstrated the possible involvement of CSCs in the bone micro-E [20], and found that bone micro-E is a key niche for CSC [14]. Generally, an activated CSC divides and produces two daughter cells, one of which is a CSC, which is usually a slowly cycling cell. The other is a cancer cell with high potential [31]. Thus, heterogeneous tumor tissue contains primarily cancer cells with high proliferation potential and a small fraction of cancer stem cells. Treatment with chemotherapeutic agents targeting cell proliferation would be expected to reduce the tumor mass by removing proliferating cells, but would also leave a significant fraction of CSCs behind.

TGF-β has been demonstrated to have dual effects on tumorigenesis, acting both as a tumor suppressor [32,33] and as a tumor promoter [34,35,36]. It is now known that this dual role is context-dependent: TGF-β is a tumor suppressor in the early stages of tumorigenesis, and enhances tumor cell survival and invasive behavior in the later stages [37,38]. TGF-β is an attractive target for therapeutic intervention. It has been published that suppressive effects of neutralizing antibody 1D11 has suppressive effects on human breast cancer cell line MDA-MB-231 [39], and that SD-208, an inhibitor of TGF-β receptor 1 kinase (TβR1-Ki) suppressed the development of melanoma bone metastasis [40] in the series of an in vivo study. We previously demonstrated that the treatment with 1D11 significantly suppressed the tumor volume and cell proliferation of mouse mammary tumor cells [17], and that treatment with ASO suppressed the tumor volume and cell proliferation of rat prostate cancer in the bone micro-E [16]. In the present study, we observed that treatment with TβR1-Ki (R1-Ki) reduced the tumor volume and suppressed cell proliferation in the bone micro-E in the implanted mouse mammary tumor cells, but R1-Ki did not induce apoptosis in the tumor. Interestingly, these suppressive effects were observed only in the bone micro-E, and not in the subQ micro-E in our previous studies [16,17], and our present study confirmed the effects caused by inhibition of TGF-β signaling only in the bone micro-E and not in the subQ micro-E. Therefore, the suppressive effects on tumor volume and tumor cell proliferation caused by inhibition of TGF-β signaling are specific to cancer cells with high proliferative potential in the bone micro-E.

In a preliminary study, treatment with chemotherapeutic drug docetaxel did not increase the pathological size of the necrotic area of a rat prostate tumor in the bone micro-E, although this treatment did increase the necrosis in the subQ micro-E, suggesting that the tumor in the bone micro-E is resistant to this chemotherapeutic agent. In the present study, we evaluated the pathological size of the necrotic area and the induction of apoptosis in the tumor after R1-Ki treatment, and we found that treatment with R1-Ki did not increase the pathological size of the necrotic area in the tumor or the induction of apoptosis in the tumor in either the bone or subQ micro-E. Similar effects were observed on the size of the necrotic area of a mouse mammary tumor in the bone micro-E after treatment with human osteoclastogenesis inhibitory factor [14].

However, R1-Ki had suppressive effects on the induction of CSCs in the bone micro-E. One of the possible reasons was that treatment with R1-Ki was not cytotoxic to the tumor cells and did not induce necrosis in the tumor is that necrosis would be induced when the tumor cells were injured by a treatment, such as chemotherapeutic agents or radiation. Another reason was that fraction of CSCs is small in heterogenous tumor tissue.

Tumor volume in vivo, cell proliferation, the size of the necrotic area in the tumor, and apoptosis in the tumor can be evaluated as a result of removing the tumor cells with high proliferative potential in the heterogeneous tumor tissue. However, these treatments may not be sufficient, because the treatments that substantially reduce the tumor mass by removing proliferating cells do not significantly prolonged the overall survival [41,42,43]. Therefore, we examined the effects of R1-Ki on the induction of CSC in the bone micro-E because CSCs are usually slowly cycling cells, and are thus insensitive to these treatments. In our previous study, we demonstrated that the number of tumor cells positive for SOX2 or CD166 in the bone micro-E were significantly higher than those in the subQ micro-E [14], and we confirmed the evidence in this study. SOX2 is a transcription factor that maintains the pluripotent properties of stem cells [44], and it is highly expressed in the cancer stem cells of breast [45,46], prostate [47], lung [48], uterine cervix [49], ovarian [50], and bladder [51] cancers. CD166 is a transmembrane glycoprotein, and was demonstrated to be a cancer stem cell marker for head and neck squamous cell carcinoma [52] and colorectal cancer [53]. These results suggest that the bone micro-E is a key niche of cancer stem cells, and we explored the possible involvement of TGF-β on the induction of cancer stem cell in the bone micro-E, and found that inhibition of TGF-β signaling suppressed the induction of cancer stem cells in the bone micro-E.

TGF-β has been shown to be involved in the self-renewal, differentiation, and survival of the CSCs of prostate [54], breast [55], and lung [56] cancers. The TGF-β-SMAD signaling is involved in modulating the cancer stem cell-like properties of CD44-positive gastric cancer cells [57], hepatocellular carcinoma cells [58], and cervical cancer cells [59]. In this study we found that reduction of TGF-β-SMAD signaling significantly reduced the CSCs in the bone micro-E, but not in the subQ micro-E. Taken together, the bone micro-E is a key niche of CSCs, and TGF-β SMAD signaling is involved in the induction of CSCs.

The MAPK/ERK signaling pathway is a major determinant in the control of diverse cellular processes such as proliferation, differentiation, and survival [60]. Its role in the regulation of cell proliferation has been widely described [61,62,63]. In this study, we demonstrated that tumor cell proliferation was increased in the bone micro-E, where the expression of pERK1/2 was up-regulated. Treatment with R1-Ki down regulated the expression of pERK1/2 and significantly reduced cell proliferation in the bone micro-E, where the expression was down-regulated. These results indicate that TGF-β activation of the MAPK/ERK pathway is the underlying mechanism of cell proliferation in the bone micro-E, which is consistent with the known role of TGF-β in the activation of the MAPK/ERK signaling pathway [64,65,66].

AKT kinases are involved in a number of important cellular processes, including cell proliferation, survival [67], enhancing cell growth [68], and inhibiting apoptosis [69]. The PI3K/AKT pathway is activated by TGF-β [70,71,72,73], and this signaling pathway has also been demonstrated to enhance bone metastasis of prostate cancer [74], renal cell carcinoma [75], and bladder cancer [76]. In this study, we demonstrated that tumor cell proliferation was increased in the bone micro-E, where the expression of p-AKT was up-regulated. Treatment with R1-Ki down-regulated the expression of p-AKT and reduced cell proliferation in the bone micro-E. Taken together, these results suggest that TGF-β signaling via the ERK1/2 and AKT pathway is involved in tumor cell proliferation in the bone micro-E.

BMP is a group of growth factors that belongs to the TGF-β family, and is involved in promoting the bone metastasis of prostate [77] and breast [78] cancers. SMAD1/5/8 are downstream molecules involved in BMP signaling [79,80]. We detected only very low levels of SMAD 1/5 in the bone and subQ micro-Es, and we did not detect p-SMAD1/5 either micro-E. Those results suggest that the BMP signaling is not involved in the tumor growth in the bone micro-E in our model.

In summary, after confirming that treatment with R1-Ki suppressed TGF-β signaling in vivo, we found that TGF-β signaling is involved in the increase of tumor volume, and was involved in tumor cell proliferation, the suppression of TGF-β signaling, and did not induce necrosis or apoptosis in the proliferating cancer cells in the bone micro-E. We also found that the bone micro-E is a key niche of CSC, and TGF-β SMAD signaling was involved in the induction of CSC in the bone micro-E. TGF-β activation of the MAPK/ERK and AKT pathway is the underlying mechanism of tumor cell proliferation in the bone micro-E. BMP signaling, however, is not involved in the tumor growth in the bone micro-E generated in our model. Our results also indicate that it is critically important, when evaluating the responses to chemotherapeutic agents on the heterogeneous tumor tissue in vivo, to monitor both proliferating cancer cells and cancer stem cells in different tumor microenvironments.

## 4. Materials and Methods

### 4.1. Tumor Cell Lines and Tissue Preparation

The murine mammary tumor cell line (Cl66M2) [24,81,82] was maintained in DMEM (Cellgro, Herndon, VA, USA) with 5% fetal bovine serum (FBS) supplemented with gentamycin (complete medium) at 37 °C in a humidified atmosphere containing 5% CO_2_. A total of 1 × 10^5^ of Cl66M2 cells, mixed with growth factor-reduced Matrigel (BD Biosciences, San Jose, CA, USA), were implanted into two locations, under the dorsal skin flap over the calvaria and into the subcutis of the back of each of female BALB/c mice. To address the contribution of TGF-β signaling to tumor growth in the bone and subQ micro-Es, after transplantation of Cl66M2 cells onto the calvaria and into the subcutis of BALB/c mice, the mice were treated with a TGF-β R1 kinase inhibitor (3-(pyridine-2-yl)-4-(4-quinonyl)-1H-Hpyrazole) (EMD Biosciences) at a dose of 10 mg/kg body weight. Tumor growth was monitored twice a week. Mice were killed and necropsied for histological examination at 4 weeks post-implantation. Cranial and subQ tumors were removed and then divided in half, with one half used for histology (hematoxylin and eosin: H&E staining) and the other half of the tumors were then flash frozen for protein analysis and RNA extraction. For histological examination, tissues were fixed with periodate-lysine-paraformaldehyde (PLP) at 4 °C for 48 h. The tissues were then transferred into a decalcification solution (15% ethylene diamine tetra acetate with glycerol, pH 7.4–7.5) for 4 weeks. The tissue was then embedded in paraffin and processed for further analysis.

### 4.2. Immunohistochemistry

For the in vivo detection of Ki-67, cleaved caspase 3, SOX2, CD166, and tumor sections were evaluated by IHC. For the IHC studies, the following diluted primary antibodies were used: cleaved caspase 3 (1:100, Cell Signal Technology Japan, Danvers, MA, USA), SOX2 (1:50, Cell Signal Technology Japan, Danvers, MA, USA), CD166 (1:50, Cell Signal Technology Japan, Danvers, MA, USA), and Ki-67 (1:100, AB-5656, USA). The entire IHC investigation was carried out using an automatic IHC machine, Leica Bond-max (Leica Microsystems, Tokyo, Japan) according to the manufacturer’s instructions. Tartrate resistant acid phosphatase (TRAP) assays were performed to detect activated osteoclasts in vivo, according to the manufacturer’s instructions (Sigma Chemicals, St Louis, MO, USA). For quantitative analyses, immune-stained sections were examined under a light microscope. The numbers of cells/nuclei positive for Ki-67, cleaved caspase 3, SOX2, and CD166 were assessed at a magnification of 400× for each lesion. Approximately 8000 cells/nuclei per tumor were counted.

### 4.3. Definition of Tumor Bone Micro-E and Tumor subQ’s Micro-E.

To define the bone micro-E, we examined the IHC analysis of p-SMAD2, because p-SMAD2 is an indicator of TGF-β signal transduction. We found that the p-SMAD2 positive cells were mainly observed in a range of 2 mm around the tumor bone interface on the cranial bone. Therefore, the area within a range of 2 mm around the tumor bone interface on the cranial bone was defined as tumor bone micro-E in this study. Accordingly, we choose the subQ micro-E, the area in a range of 2 mm at the tumor subQ interface of the subcutaneous tumor, as the control micro-E in this study.

### 4.4. Statistical Analysis

For in vivo data, statistical analysis was performed using the Kruskal–Wallis and Bonferroni–Dunn’s multiple comparison tests. In vitro data are presented as means ± standard deviations. The statistical significance of in vitro findings was analyzed using a two-tailed Student’s *t*-test and Bonferroni–Dunn’s multiple comparison tests. A value of *p* < 0.05 was considered significant. The Spearman’s rank correlation test was used to determine dose-response correlation.

## Figures and Tables

**Figure 1 ijms-20-05117-f001:**
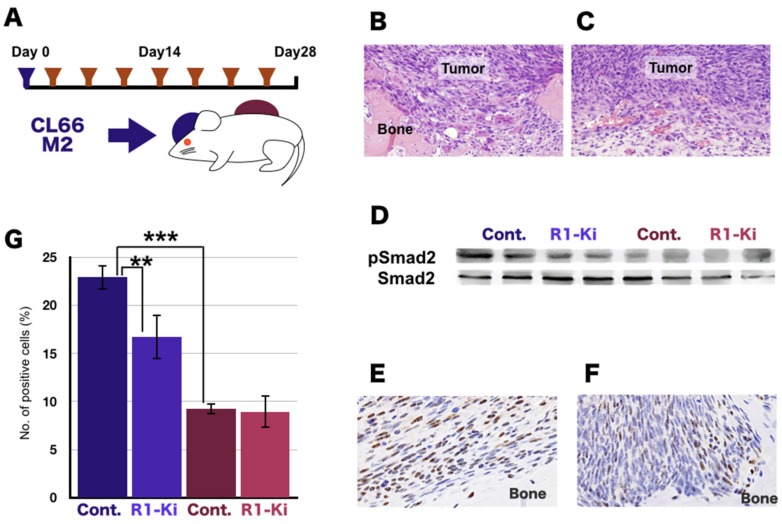
Suppression of TGF-β signaling by TGF-β receptor kinase inhibitor in the bone microenvironment (micro-E). (**A**) Experimental design: mouse mammary tumor cell line was implanted into two different sites in female mice, under the dorsal skin flap over the calvaria and into a subcutaneous lesion. Then the mice were injected intraperitoneously seven times with a TGF-β receptor 1 kinase inhibitor (R1-Ki) over the course of the experimental period. (**B**) Typical examples of tumor tissue in the bone micro-E. Strong osteolysis associated with induction of numerous osteoclasts was observed in the bone micro-E. Magnification: 600×. (**C**) Typical example of the tumor cells in the subcutaneous (subQ) micro-E; growth of the tumor cells was observed with micro vessel invasion. Magnification: 600×. (**D**) Western blot analysis of the expression of phosphorylated-SMAD2, and SMAD2. (**E**, **F**) IHC staining of pSMAD2: Many positive cells were observed in the control group (**E**) and a smaller number of positive cells were in the R1-Ki treatment group in the bone micro-E (**F**). Magnification: 600×. (**G**) Quantitative analysis of p-SMAD2 positive cells. **, ***: *p* < 0.01, *p* < 0.001.

**Figure 2 ijms-20-05117-f002:**
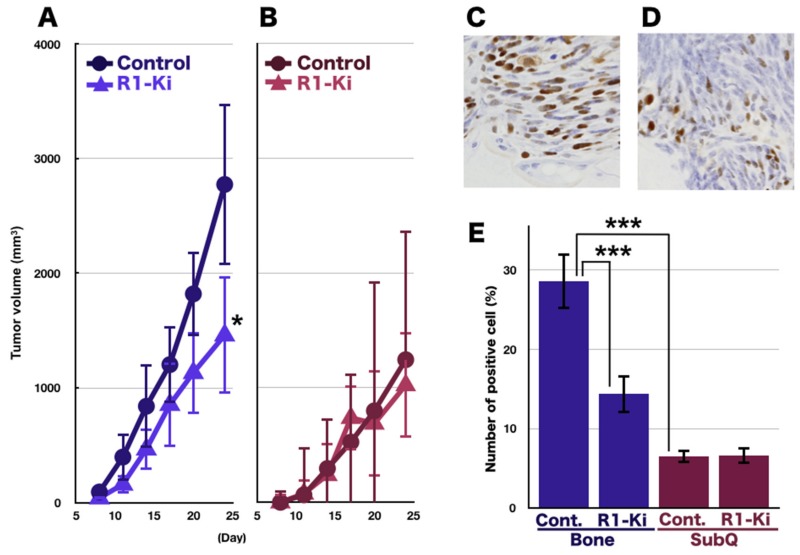
The effects of TGF-β on tumor growth and cell proliferation. (**A**) Tumor size in the bone micro-E: Faster tumor growth in the control group and slower growth in the R1-Ki treatment group. A significant difference in tumor size was observed at Day 24. (**B**) Tumor size in the subQ micro-E: Slow tumor growth regardless of treatment. (**C**,**D**) Many Ki-67 positive cells in the control group in the bone micro-E (**C**), and fewer positive cells in the subQ micro-E (**D**). Magnification: 600×. (**E**) Significant suppression of Ki-67 positive cells by the treatment with R1-Ki in the bone micro-E but not in the subQ micro-E. *, ***: *p* < 0.05, *p* < 0.001.

**Figure 3 ijms-20-05117-f003:**
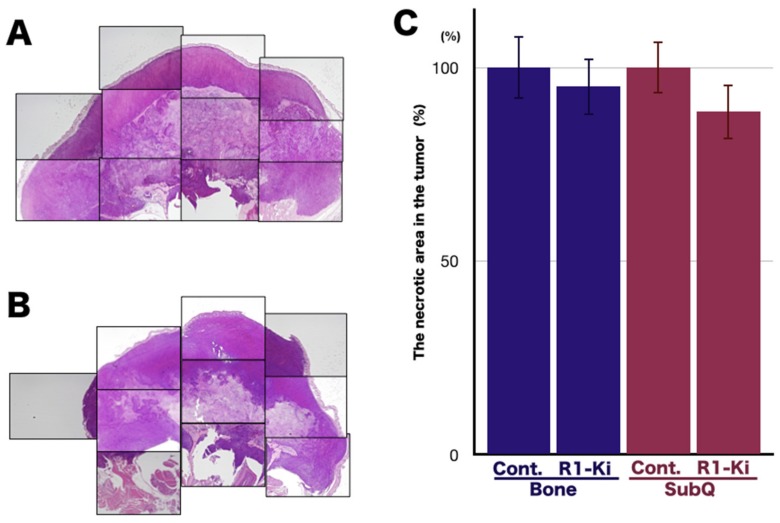
The effect of TGF-β signaling on the induction of necrotic area. (**A**,**B**): Evaluation of the necrotic area in the tumors by microscopic analysis and image analyzer in the control group (**A**) and R1-Ki treatment group (**B**). Magnification: 2×. **C**: Quantitative analysis of the necrotic area: R1-Ki had no effect on the necrotic area in either the bone or subQ micro-E.

**Figure 4 ijms-20-05117-f004:**
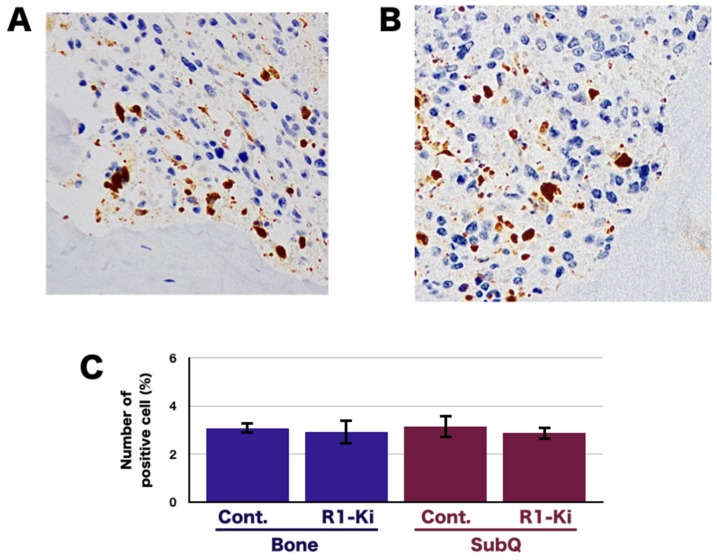
The effect of TGF-β signaling on the induction of necrotic area and apoptosis. (**A**,**B**) IHC staining of cleaved caspase 3 in the control group (**A**) and in the R1-Ki treatment group (**B**). (**C**) Quantitative analysis of cells positive for cleaved caspase 3.

**Figure 5 ijms-20-05117-f005:**
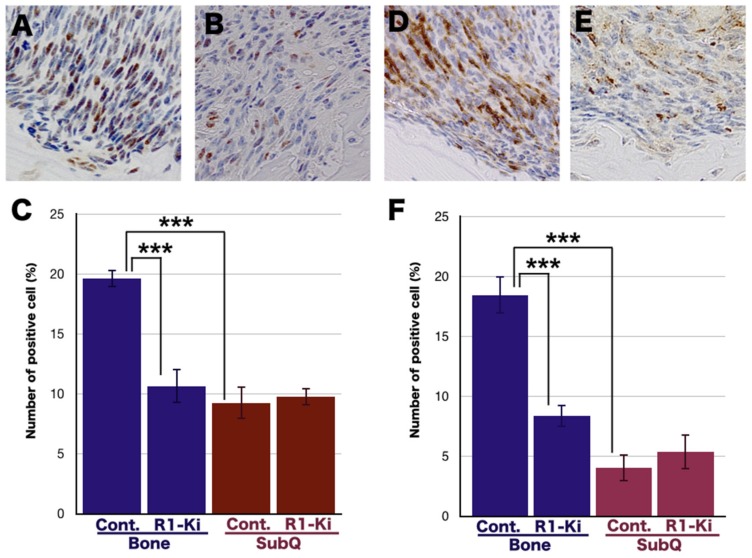
The effect of TGF-β signaling on the induction of CSC in the bone micro-E. (**A**,**B**) IHC study of SOX2 staining in the bone micro-E: A large number of SOX2 positive cells can be seen in the control group (**A**) and fewer positive cells in the R1-Ki treatment group (**B**). Magnification: 600×. (**C**) Quantitative analysis: R1-Ki treatment resulted in a significant suppression of SOX2 positive cells in the bone micro-E but not in the subQ micro-E. (**D**,**E**) IHC study of CD166 staining in the bone micro-E.; A large number of positive cells can be seen in the control group (**D**) and fewer positive cells in the R1-Ki treatment group (**E**). Magnification: 600×. (**F**) Quantitative analysis; R1-Ki treatment resulted in a significant suppression of CD166 positive cells in the bone micro-E, but not in the subQ micro-E. ***: *p* < 0.001.

**Figure 6 ijms-20-05117-f006:**
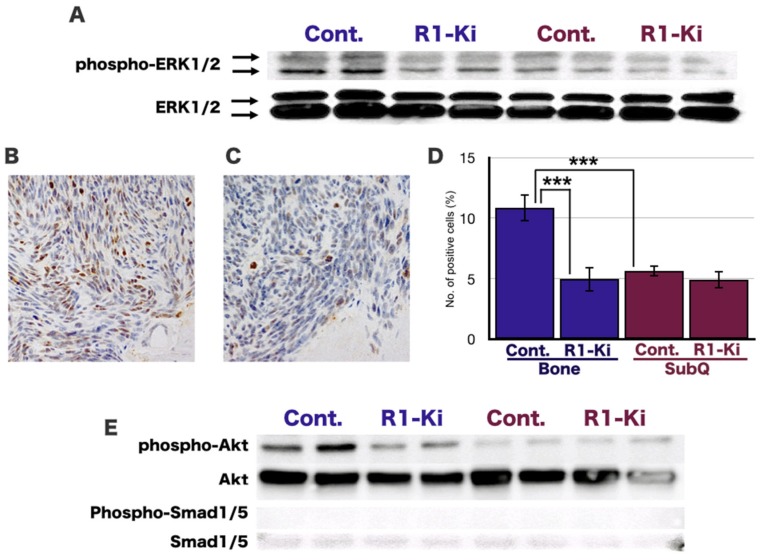
The involvement of ERK1/2 and AKT, and BMP signaling, in the bone and subQ micro-Es. **A**: Western blot analysis: The expression of p-ERK1/2 was relatively strong in the bone micro-E and weak in the subQ micro-E. The expression of ERK1/2 was similar in the bone and subQ micro-Es regardless of treatment (**A**). (**B**,**C**) IHC study of p-ERK1/2 staining in the bone micro-E: A large number of p-ERK1/2 positive cells can be seen in the control group (**B**) and fewer positive cells in the R1-Ki treatment group (**C**). Magnification: 400×. (**D**) Quantitative analysis of p-ERK1/2 positive cells: R1-Ki treatment significantly reduced the number of positive cells in the bone micro-E but not in the subQ micro-E. (**E**) Expression of phosphorylated AKT and AKT: Strong expression of phosphorylated AKT was observed of the control group and weaker expression of the R1-Ki treatment group in the bone micro-E. Weak expression of p-AKT was observed in both the control and R1-Ki treated groups in the subQ micro-E. The expression of AKT was similar in the bone and subQ micro-Es regardless of treatment. (**E**) Expression of SMAD1/5: The expression of SMAD1/5 was only slightly detected, and the expression of p-SMAD1/5 was not detected in either the bone or subQ micro-E regardless of treatment. ***: *p* < 0.001.

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
