# Peer review of "The Effects of TGF-β Signaling on Cancer Cells and Cancer Stem Cells in the Bone Microenvironment"

_ijms, 2019, doi:10.3390/ijms20205117_

Round 1

Reviewer 1 Report

I have the following concerns and suggestions regarding this manuscript;

1) Moderate English language editing is required.

2) The legend for Fig 2C, D does not fit well with the description in the text (lines 130-132). In particular,  2 different markers were mentioned to assess the proliferation of tumor cells: PCNA (in the figure legend) and Ki-67 (in the text).

3)  It is not clear what kind of difference the authors are trying to show in Fig 2E for subQ-micro E lesions.  It was mentioned in the text and shown in figure 2E that TGF-beta inhibitor did not have an impact on the proliferative index... Similar concerns are for the Fig 5C, F and Fig.6D. It looks like the authors are trying to use the bone micro-E for the control for both types of lesions. if so, this is not correct...   

4) The authors have to explain the high numbers of apoptotic ( e.g. caspase-3 -positive) cells in control (non-treated) mice (shown in Fig 4A). IHC-data does not fit well with the numbers that were shown on the graphs.  

5) Generally, PTEN has not used as a marker of apoptosis and therefore it is not correct to use it in this context (as shown in the lines 150-160 and Fig 4C and D). 

6) It is not clear what was shown on Fig 4C and F (the numbers of apoptotic cells or smth else?). The axis Y should be named and reflect IHC-data.  

Author Response

Reviewer 1

Comment 1)

Moderate English language editing is required.

Response to the comment 1)

We have English editing extensively, and our manuscript was checked by two professional English editors.

Comment 2)

The legend for Fig 2C, D does not fit well with the description in the text (lines 130-132). In particular,  2 different markers were mentioned to assess the proliferation of tumor cells: PCNA (in the figure legend) and Ki-67 (in the text).

Response to the comment 2)

We really appreciate your comment. We described “the R1-Ki treatment significantly reduced...” in the text, but Figure 2C, and D (old version) indicate “ compared to the fewer number of positive cells in the subQ micro-E (Figure 2D)”. We believe these discrepant descriptions may brought confusion to the reviewer.

To avoid such confusion, we deleted the following text, “, compared to the fewer number of positive cells in the subQ micro-E (Figure 2D)”

and, we revised the PCNA to Ki-67 in Figure legends.

Comment 3)

It is not clear what kind of difference the authors are trying to show in Fig 2E for subQ-micro E lesions.  It was mentioned in the text and shown in figure 2E that TGF-beta inhibitor did not have an impact on the proliferative index... Similar concerns are for the Fig 5C, F and Fig.6D. It looks like the authors are trying to use the bone micro-E for the control for both types of lesions. if so, this is not correct...   

Response to the comment 3)

Thank you for the comment. Similar to the comment 2), we were “trying to use the bone micro-E for the control for both type of lesions.” But Figures 1G, 2E, 5C, 5F, and 6D, (old version) looks like the subQ micro-E for the control.

Therefore, we revised Figures 1G, 2E, 5C, 5F, and 6E (new version) to demonstrate statistic difference between two groups. Accordingly, we revised Figure legends.

Figure 1G  vs Con at Control group

Figure 2E vs Con. in the bone subQ micro-E.

Figure 5C, 5F vs Con in the bone micro-E.

Figure 6D vs Con in the bone micro-E

Comment 4)

The authors have to explain the high numbers of apoptotic ( e.g. caspase-3 -positive) cells in control (non-treated) mice (shown in Fig 4A). IHC-data does not fit well with the numbers that were shown on the graphs.  

Response to the comment 4)

Thanks for your careful comment. In old Figure 4, the number of positive cells in Figure 4A looks like smaller than those in Figure 4B, although the graph (Figure 4C) indicates some difference between them.

We revised the Figure 4A(new version) by choosing better figure. We believe that IHC-data fit well with the number that were shown on the graphs in the new version of Figure 4.

Comment 5)

Generally, PTEN has not used as a marker of apoptosis and therefore it is not correct to use it in this context (as shown in the lines 150-160 and Fig 4C and D). 

Response to the Comment 5)

Thank you for the comments. According to the comments, we deleted the results of PTEN in this manuscript.

D, E: IHC staining of PTEN in the control group (D) and in the R1-Ki treatment group (E).

Comment 6)

It is not clear what was shown on Fig 4C and F (the numbers of apoptotic cells or smth else?). The axis Y should be named and reflect IHC-data.  

Response to the comment 6)

Thank you for your comments. We named the axis Y in the graph as “number of positive cells (%)” in Figure 4C (new version).

Reviewer 2 Report

Please, see the comments in the attached file.

The authors demonstrate the importance of TGFb-signalling in the bone microenvironment for the growth of cancer stem cells (CSCs). To study the role of TGF-b pathway in the tumor microenvironment at the bone interface, they injected CL66M2 mammary cells under the dorsal skin flap over the calvaria of BALB/c syngeneic mice. Moreover, they also injected the same cells subcutaneously far away from bone to study the same pathway in a different tumor microenvironment (subQ micro-E) that does not interact with bone.

In figure 1, they showed that expression of p-SMAD2 was high in the bone microenvironment (micro-E) and low in the subQ micro-E (Figure 1D). Moreover, the expression of p-SMAD2 in the bone micro-E was decreased by treatment with TGF-βRI kinase inhibitor (R1-Ki). However, the same group already reported this data in 2009, using the same TGF-βRI kinase inhibitor (even if in that case they called it ALK5-i):

Futakuchi M et al., Transforming growth factor-b signaling at the tumor–bone interface promotes mammary tumor growth and osteoclast activation, Cancer Sci. 2009

Figure 7 Transforming growth factor (TGF)-βRI kinase inhibitor reduced mammary tumor-induced osteolysis. (A) Treatment with ALK5-i inhibited tumor growth (n = 6 mice per group). (B) Significant reduction in bone destruction in ALK5-i-treated mice compared to control. (C) Significantly fewer osteoclasts were observed at the tumor–bone (TB) interface in the ALK5-i-treated mice. (D) A significant inhibition in cell proliferation at the TB interface was observed inALK5-i-treated mice as compared to control. (E) pSmad2 immunostaining in ALK5-i-treated and untreated group. The values are mean average ± standard deviation. *Significantly different from tumoralone area (*P < 0.05; **P < 0.01; ***P < 0.001)

Next, the author showed that in the bone micro-E, tumor growth was increased in the control mice compared to the R1-Ki treated mice, resulting in a significant difference in the tumor size on Day 24. However, the same group already reported this data in 2009, using the same TGF-βRI kinase inhibitor (even if in that case they called it ALK5-i):

Figure 7 Transforming growth factor (TGF)-βRI kinase inhibitor reduced mammary tumor-induced osteolysis. (A) Treatment with ALK5-i inhibited tumor growth (n = 6 mice per group). (B) Significant reduction in bone destruction in ALK5-i-treated mice compared to control. (C) Significantly fewer osteoclasts were observed at the tumor–bone (TB) interface in the ALK5-i-treated mice. (D) A significant inhibition in cell proliferation at the TB interface was observed inALK5-i-treated mice as compared to control. (E) pSmad2 immunostaining in ALK5-i-treated and untreated group. The values are mean average ± standard deviation. *Significantly different from tumoralone area (*P < 0.05; **P < 0.01; ***P < 0.001).

What is new is that they showed that the tumor grew more slowly in the subQ lesion compared to the growth in the bone lesion, and R1-128 Ki treatment did not suppress the tumor growth in the subQ micro-E. These data highlight the importance of this pathway specifically in the tumor microenvironment at the bone interface. However, this conclusion was also deducible from their previous paper, where they compared the tumor microenvironment at the bone-interface with the tumor microenvironment far away from the bone but within the same tumor. Here, they compared the subcutaneous tumor micro-E in a tumor grown close to the calvaria bone with the subcutaneous tumor micro-E of a different tumor, grown in a different area, which was not close to any bone.

Next, the authors showed that R1-Ki had no effect on the necrotic area in either the bone or subQ micro-E.

However, they showed that R1-Ki had suppressive effects on the induction of CSCs in the bone micro-E. These results indicate that the bone micro-E is a key niche for cancer stem cells, and that TGF-β is involved in the induction of CSC in the bone micro-E. Interestingly, they reported that ERK 1/2 and AKT signalling but not BMP signalling, are involved in tumor growth in the bone micro-E. However, in the western blot analysis of SMAD1/5, they should provide a positive control showing the activated p-SMAD1/5 protein.

In conclusion, the only novelty in the work is that the suppressive effects on tumor volume and tumor cell proliferation caused by inhibition of TGF-β signalling are specific to cancer cells present at bone interface. However, the authors did not provide any model of bone metastasis. A metastatic model would better resemble what happens in breast cancer patients who develop bone metastasis.

Author Response

Reviewer 2

Comment 1

In figure 1, they showed that expression of p-SMAD2 was high in the bone microenvironment (micro-E) and low in the subQ micro-E (Figure 1D). Moreover, the expression of p-SMAD2 in the bone micro-E was decreased by treatment with TGF-βRI kinase inhibitor (R1-Ki). However, the same group already reported this data in 2009, using the same TGF-βRI kinase inhibitor (even if in that case they called it ALK5-i):

Comment2) Next, the author showed that in the bone micro-E, tumor growth was increased in the control mice compared to the R1-Ki treated mice, resulting in a significant difference in the tumor size on Day 24. However, the same group already reported this data in 2009, using the same TGF-βRI kinase inhibitor (even if in that case they called it ALK5-i):

What is new is that they showed that the tumor grew more slowly in the subQ lesion compared to the growth in the bone lesion,and R1-128 Ki treatment did not suppress the tumor growth in the subQ micro-E.

However, this conclusion was also deducible from their previous paper, where they compared the tumor microenvironment at the bone-interface with the tumor microenvironment far away from the bone but within the same tumor.

Here, they compared the subcutaneous tumor micro-E in a tumor grown close to the calvaria bone with the subcutaneous tumor micro-E of a different tumor, grown in a different area, which was not close to any bone.

Response to the comment 1)and 2)

In the previous study, we examined the effects of TGF-βRI kinase inhibitor on the expression of p-SMAD2, tumor size, and cell proliferation (citation), and we compared the tumor cells at the bone-interface with those in the area far away from the bone but within the same tumor.

Because these two areas are located in the same tumor, we can not completely deny the possibility that tumor cells in the area far away from the bone are affected by TGFβ at the bone-interface.

In addition, the definition of tumor-bone interface and tumor-bone microenvironment remained obscure.

Therefore, in this study, we defined the bone microE as the area where TGFβ has influence on because the purpose of this study was the effects of TGFβ signaling in the bone microE.

To define the bone micro-E, we examined the IHC analysis of p-SMAD2 because p-SMAD2 is an indicator of TGF-β signal transduction.

We found that the p-SMAD2 positive cells were mainly observed in the area in a range of 2mm at the tumor bone interface on the cranial bone.

Therefore, the area in a range of 2mm at the tumor bone interface on the cranial bone was defined as tumor bone micro-E in this study.

Accordingly, we choose the subQ micro-E, the area in a range of 2mm at the tumor subQ interface of the subcutaneous tumor, as the control micro-E in this study.

Although the conclusion in this manuscript was deducible from the previous paper, we examined the effects of TGF-βRI kinase inhibitor of the bone micro-E comparing with the subQ micro-E based on the definition of the micro-E in this study.

Therefore, in this study, we injected the same number of Cl66M2 in the subcutaneous lesion to generate the subcutaneous tumor micro-E, which was not close to any bone as the control micro-E.

We have added the following sentences in the discussion section.

In the previous study, we examined the effects of TGF-βRI kinase inhibitor on the expression of p-SMAD2, tumor size, and cell proliferation [17], and we compared the tumor cells at the bone-interface with those in the area far away from the bone but within the same tumor. Because these two areas are located in the same tumor, we can’t completely deny the possibility that tumor cells in the area far away from the bone are affected by TGFβ at the bone-interface. In addition, the definition of tumor-bone interface and tumor-bone micro-E remained obscure. Therefore, in this study, we defined the bone microE as the area where TGFβ has influence on because the purpose of this study was the effects of TGFβ signaling in the bone micro-E.

We have added the following sentences in the materials and methods section.

Definition of bone micro-E in this study

To define the bone micro-E, we examined the IHC analysis of p-SMAD2 because p-SMAD2 is an indicator of TGF-β signal transduction. We found that the p-SMAD2 positive cells were mainly observed in the area in a range of 2mm at the tumor bone interface on the cranial bone. Therefore, the area in a range of 2mm at the tumor bone interface on the cranial bone was defined as tumor bone micro-E in this study.

Comment 3)

Next, the authors showed that R1-Ki had no effect on the necrotic area in either the bone or subQ micro-E. However, they showed that R1-Ki had suppressive effects on the induction of CSCs in the bone micro-E. These results indicate that the bone micro-E is a key niche for cancer stem cells, and that TGF-β is involved in the induction of CSC in the bone micro-E.

Response to the comment 3)

Thank you for this comment. We showed that R1-Ki had no effect on the necrotic area in either the bone or subQ micro-E although R1-Ki had suppressive effects on the induction of CSCs in the bone micro-E.

One of the reasons was that the treatment with R1-Ki was not cytotoxic to the tumor cells and did not induce necrosis in the tumor, which we described in the discussion section.

Another possible reason was that fraction of CSCs is small in the heterogenous tumor tissue.

Therefore, we added the following sentences in the discussion section.

Similar effects were observed on the size of the necrotic area of a mouse mammary tumor in the bone micro-E after treatment with human osteoclastogenesis inhibitory factor [14].

However, R1-Ki had suppressive effects on the induction of CSCs in the bone micro-E. One of the possible reasons was that treatment with R1-Ki was not cytotoxic to the tumor cells and did not induce necrosis in the tumor because necrosis would be induced when the tumor cells were injured by a treatment, such as chemotherapeutic agents or radiation.Another reason was that fraction of CSCs is small in the heterogenous tumor tissue.

Comment 4

Interestingly, they reported that ERK 1/2 and AKT signalling but not BMP signalling, are involved in tumor growth in the bone micro-E. However, in the western blot analysis of SMAD1/5, they should provide a positive control showing the activated p-SMAD1/5 protein.

Response to the Comment 4)

We added the western blotting analysis of the positive control of p-SMAD1/5 and SMAD1/5 in Figure 6F.

We added the following sentence in the result section and Figure legends.

Expression of p-SMAD1/5 and SMAD1/5 was confirmed by Western blot analysis of extracts from BMP-4 treated HeLa cells and untreated HeLa cells, respectively.

Comment 5

In conclusion, the only novelty in the work is that the suppressive effects on tumor volume and tumor cell proliferation caused by inhibition of TGF-β signalling are specific to cancer cells present at bone interface. However, the authors did not provide any model of bone metastasis.  A metastatic model would better resemble what happens in breast cancer patients who develop bone metastasis.

Response to the comment 5)

We added the following sentences in the discussion section.

Bone metastasis is multiple processes, that involves involve invasion, survival in the circulation, arrest in bone, and re-growth in the bone micro-E [21].Ideally, an in vivomodel of bone metastases should reflect the natural course of the disease and accurately simulate the disease progression commonly observed in advanced cancer patients.However, such animal model that fully reflect the biology of bone metastasis is a major obstacle to develop therapies [22-24] because each process of bone metastasis involves independent mechanism. We have developed a rat bone invasion model to observe the growth of prostate or breast cancer cells with osteoblastic / osteolytic lesions in the bone micro-E [14, 16-19]. Although our models do not recapitulate the entire metastatic process that primary tumor cells in prostate or breast move to the typical location in the bone, our model does represent how metastatic tumor cells interact with the bone stromal cells in the bone microenvironment. Using our model, we identified a unique pattern of gene expression that was up-regulated at the TB interface, including genes such as RANKL, which are known to be involved in bone metastasis [17, 18]. Therefore, our model provides an exciting opportunity to elucidate the molecular mechanisms and to develop therapeutic targets underlying tumor stromal interaction in the bone micro-E.

Round 2

Reviewer 1 Report

In general, the authors responded properly to the suggestions and comments.

However, the authors should carefully revise all the manuscript and fix some technical issues before the manuscript might be considered acceptable for publication. 

1) For example, according to comment 5, the authors deleted PTEN data from the figure legend. However, PTEN data is still present (!) in the text (lines 151, 155-159). PTEN info should be also removed from the Material and methods section  (lines 384, 387, 394).

2) The Y-axis in Fig 3C should be renamed to the "number of apoptotic cells (%)"

3) Fig 2E still contains PCNA data (line 141). Similarly, in the Material and methods section, PCNA is still present (lines 384, 389, 394)

Author Response

Reviewer 1

In general, the authors responded properly to the suggestions and comments.

However, the authors should carefully revise all the manuscript and fix some technical issues before the manuscript might be considered acceptable for publication. 

Comment 1)

For example, according to comment 5, the authors deleted PTEN data from the figure legend. However, PTEN data is still present (!) in the text (lines 151, 155-159). PTEN info should be also removed from the Material and methods section  (lines 384, 387, 394).

Response to the comment 1)

We apologize our careless mistakes. We have removed all the information related to PTEN study.

Comment 2)

The Y-axis in Fig 3C should be renamed to the "number of apoptotic cells (%)"

Response to the comment 2)

Thank you for the comment. Because “The necrotic area in the tumors was analyzed by microscopic analysis and image analyzer (Figures 3A, 3B).”,the Y-axis in Fig 3C should be “The necrotic area in the tumor”.

Comment 3)

Fig 2E still contains PCNA data (line 141). Similarly, in the Material and methods section, PCNA is still present (lines 384, 389, 394)

Response to the comment 3)

We apologize our careless mistakes. We have removed all the information related to PTEN study.

Because we evaluated the cell proliferation by Ki-67 but not PCNA, we have changed PCNA to Ki-67 in the text.
